# Opioid and gabapentinoid prescriptions in England from 2015 to 2020

Yixue Xia[1], Patrice Forget [1,2,3]*

**1** Epidemiology group, Institute of Applied Health Sciences, School of Medicine, Medical Sciences and Nutrition, University of Aberdeen, Aberdeen, United Kingdom, **2** Anaesthesia department, NHS Grampian, Aberdeen, United Kingdom, **3** The European Society of Anaesthesiology and Intensive Care (ESAIC), Pain AND Opioids after Surgery (PANDOS), Research Group, Brussels, Belgium

* forgetpatrice@yahoo.fr

**Data Availability Statement:** Data are freely available on www.openprescribing.net and www.ons.gov.uk. Please find here the respective URLs: https://openprescribing.net/analyse/#org=regional_team&numIds=4.7,0408010G0AA,0408010AE AA&denom=nothing&selectedTab=summary

## Abstract

### Purpose

Concerns gradually arose about misuse of gabapentinoids (gabapentin and pregabalin), especially when used in combination with opioids. Because it can be a driver of usage, trends in prescribing habits may be interesting to analyse. The aim of this study is to examine the evolution of prescriptions of opioids and gabapentinoids in England from 2015 to 2020 at a regional level.

### Methods

This study included data from April 2015 to February 2020, focusing on prescribing data, extracted the OpenPrescribing database. We described the evolution of the prescriptions of opioids and gabapentinoids and calculated their ratios for each month. We used Analyses of Variance (ANOVAs) to compare data between and within regions (over time).

### Results

During this period, opioid prescriptions remained stable (from -3.3% to +2.2%/year) and increased for gabapentinoids generally (from +1.5% to +2.2%). The ratio between gabapentinoid to opioid prescriptions increased by more than 20% in 2020 compared to 2015, variably between regions (F(6,406) = [120.2]; P<0.001; LSD Test: P<0.001; ANOVA for repeated measures: P<0.05). In 2019, a decline in the ratio occurred in all regions, but only persisting in the London commissioning region in 2020 (-14.4% in comparison with 2018, 95%CI: -12.8 to -16.3).

### Conclusions

Gabapentinoids are increasingly prescribed in England. The ratio of gabapentinoid to opioid prescriptions in England increased from 2015 to 2020. The reclassification of gabapentinoids as controlled drugs, in 2019, may have been associated with a significant reduction, although larger prescribers may have been less influenced.

https://www.ons.gov.uk/peoplepopulationand
community/birthsdeathsandmarriages/deaths/
datasets/deathsregisteredinenglandandwalesse
riesdrreferencetables.

**Funding:** The author(s) received no specific
funding for this work.

**Competing interests:** The authors have declared
that no competing interests exist.

**Abbreviations:** ANOVA, Analysis of Variance; CCG,
Clinical Commissioning Group; GP, General
Practitioner; NHS, National Health Service; SD,
Standard deviation; STROBE, Strengthening
Reporting in Observational Studies.

## Background

Opioid analgesics include alkaloids extracted from opium and synthetic analogues that interact
with specific central receptors. Usually, opioids are prescribed to relieve acute pain or pain at
the end of life [1]. But little evidence supports a lasting effect on chronic pain. However, the
use of opioids is not completely safe. Adverse effects of opioids are related to various factors,
such as individual differences, dosage, and drug interactions, which are not specific to drug
type and route of administration [2].

Gabapentinoids include pregabalin and gabapentin, both of which were initially cleared for
seizures. With opioid use disorder becoming an international public health problem, doctors
and patients are looking for alternatives. As a result, more prescriptions for gabapentinoids
may have been issued. Even if largely uncertain, it was hypothesised by some that gabapenti-
noids may help reduce opioids, prevent opioid tolerance, improve the quality of opioid analge-
sic therapy, and treat anxiety [3]. However, these hypotheses have been, at least partially,
rejected [4, 5]. Moreover, in high doses, patients may experience euphoria with gabapenti-
noids, and withdrawal after abruptly stopping use, including seizures [5–7]. In recent years,
the gabapentinoid use disorder has developed rapidly and has gradually become a recognized
problem around the world. In patients with substance use disorders, especially those involving
opioid use, the abuse may even be more serious [8] and the United Kingdom (UK) classified it
as a controlled substance in April 2019 [9]. Additionally, the combined use of gabapentinoids
and opioids appears to be associated with specific risks. In patients being prescribed opioids
and gabapentinoids concomitantly, there is a substantial increase in the risk of opioid-related
death [10, 11]. This concomitant use has been accused of being responsible of differences in
opioid-related death rates, between different parts of the UK [12]. A similar debate occurred
regarding the concomitant use of benzodiazepines and gabapentinoids [13]. Co-prescribing of
benzodiazepines is well described, but there is little description of the co-evolution of the pre-
scribing of opioids and gabapentinoids, except for a recent publication in Scotland [10]. This
highlights the need to study regional and national differences.

The aim of this study is to examine the evolution of prescriptions of opioids and gabapenti-
noids in England from 2015 to 2020 at a regional level.

## Methods

This report is written according to the Strengthening Reporting in Observational Studies
(STROBE) guidelines.

### Data sources and preparation

We focused on monthly practice -level data in England from April 2015 to March 2020 and
aggregated it at a regional level. In each National Health Service (NHS) Primary Care prescrib-
ing organization in the UK, the prescribing data set monthly published by NHS Digital counts
for each different drug and dose, describing the number of prescriptions and the total cost.
These data come from institutions such as community pharmacies and contain all drugs that
have been assigned.

We extracted prescription information from the open OpenPrescribing database for analy-
sis [15]. OpenPrescribing is an online service launched in 2016. Prescribing information is
sourced from NHS Digital publishing NHS Business Services Administration monthly and
annual prescription data sets and static prescribing trend reports. All data is grouped by drug
name and any name available in multiple formulas is combined. OpenPrescribing collects
information coming from the following sources: Medications codes and names are also from
the NHS Business Service Authority's Information Portal. Clinical Commissioning Groups

(CCGs) and practice prescribing settings, are from NHS Digital's data downloads (epraccur. csv), used under the terms of the Open Government Licence. CCG names and codes and CCG geographic boundaries are from the Office for National Statistics (ONS) ([www.ons.gov.uk](www.ons.gov.uk)). Practice locations are based on data from NHS Digital/ONS.

### Data extraction and classification

We extracted prescription data (opioid analgesics, section 4.7.2 of the formularium) (for a full list, see [https://openprescribing.net/bnf/040702](https://openprescribing.net/bnf/040702)/; pregabalin, code 0408010AE; and gabapentin, code 0408010G0) as.csv files. All orders (from April 2015 to February 2020) were compiled according to different regions of England. Capital expenditure was counted per year, covering the annual total of pregabalin and gabapentin prescriptions across all NHS England regional teams.

### Statistical analysis

After compiling the prescriptions data (opioids and gabapentinoids), we calculated the ratios of gabapentinoid (gabapentin and pregabalin) to opioid prescriptions. We displayed these using a line graph with their 95% confidence intervals, calculated for each month, on a rolling year.

The prescription ratios were analysed, after checking the distribution normality using graphical methods, using a one-way Analysis of Variance (ANOVA) to compare prescription counts, and ratios between gabapentinoid and opioid prescriptions, between the different regions, and ANOVAs for repeated measures to compare each region over time. IBM SPSS (SPSS Statistics for Windows, version 25 (SPSS Inc., Chicago, Ill., USA) was used for all the comparison. A P-value <0.05 was considered statistically significant.

### Ethics approval

This study does not require patient consent or ethical approval as the data used is publicly available, anonymous and aggregated at source.

## Results

### Evolution of opioid prescription in England

Between April 2015 and February 2020, the number of opioid prescriptions in regions of England generally remained stable, ranging from -3.3% to +2.2% of change every year (Fig 1, Table 1).

### Evolution of gabapentinoid prescription in England

From April 2015 to February 2020, the number of prescriptions of gabapentin and pregabalin in England continuously increased, ranging from +1.5% to 11.9% every year (Fig 2, Table 1).

In early 2019, all regions simultaneously showed a decline, but only persistent in the London commissioning region, falling and remaining 12.3% lower than before.

### Evolution of the ratio between gabapentinoid and opioid prescription in England, between and within regions

To study conjointly the evolution of gabapentinoid and opioid prescription, we calculated the ratio of gabapentin and pregabalin to opioids. A one-way ANOVA was performed to compare

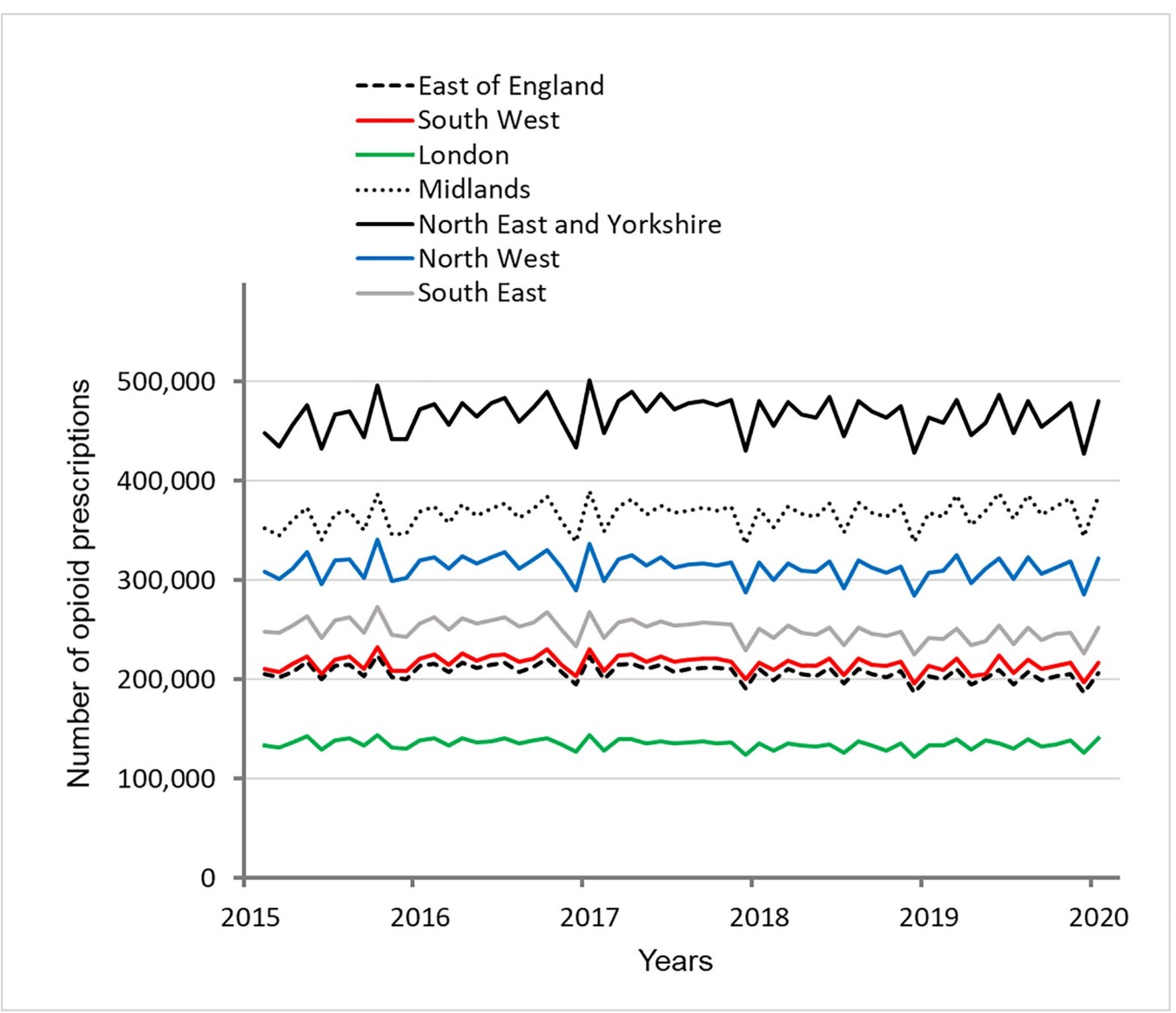

**Fig 1. Number of prescriptions for opioid analgesics in each region of England.**

the ratios and showed statistically significant difference between regions (F(6,406) = [120.2]; P<0.001; LSD Test: P<0.001).

Between April 2015 and February 2020, the evolution of the ratio increased overall, with each region experiencing an increase of more than 20% in 2020 compared to 2015 (P<0.05) (Fig 3, Table 1). In 2019, the decrease in gabapentinoid prescriptions was associated with a variably distributed decline in the ratio between gabapentinoid and opioid prescriptions, significant in all regions (ANOVA for repeated measures within regions: P<0.001) but only persisting in the London commissioning region in 2020 (-14.4% in comparison with 2018; 95%CI: -12.8 to -16.3). Costs were explored while considering all prescriptions as a whole, even if a pharmaco-economic study is beyond the purpose of this work. Costs linked to prescriptions increased rapidly from 2015 to 2019, and the London commissioning region increased the most.

**Table 1. Mean monthly prescription count (averaged on the yearly basis) for opioids and gabapentinoids, and their ratio (with 95% confidence intervals– 95%CI–calculated, for each month, on a rolling year) in each region of England.**

| | Prescriptions (monthly average) | East of England | South West | London | Midlands | North East, Yorkshire | North West | South East |
|---|---|---|---|---|---|---|---|---|
| **2015** | Opioids | 209813 | 216462 | 136637 | 360289 | 457997 | 314235 | 255071 |
| | Gabapentinoids | 99462 | 87962 | 88537 | 160010 | 185128 | 157301 | 112926 |
| | Gabapentinoid/Opioid ratio (95%CI) | 0.474 (0.461 to 0.493) | 0.406 (0.393 to 0.419) | 0.648 (0.625 to 0.671) | 0.444 (0.428 to 0.460) | 0.404 (0.392 to 0.416) | 0.500 (0.483 to 0.518) | 0.443 (0.425 to 0.460) |
| **2016** | Opioids | 211687 | 220083 | 137142 | 366760 | 468079 | 317519 | 256319 |
| | Gabapentinoids | 111169 | 98127 | 100452 | 177724 | 206198 | 174201 | 125536 |
| | Gabapentinoid/Opioid ratio (95%CI) | 0.525 (0.515 to 0.540) | 0.446 (0.436 to 0.455) | 0.732 (0.714 to 0.750) | 0.484 (0.472 to 0.497) | 0.440 (0.431 to 0.449) | 0.548 (0.535 to 0.562) | 0.489 (0.476 to 0.503) |
| **2017** | Opioids | 210213 | 218992 | 135921 | 367791 | 472951 | 315138 | 253914 |
| | Gabapentinoids | 123264 | 109317 | 113120 | 197516 | 229733 | 192258 | 139521 |
| | Gabapentinoid/Opioid ratio (95%CI) | 0.586 (0.581 to 0.596) | 0.499 (0.494 to 0.504) | 0.832 (0.815 to 0.849) | 0.537 (0.528 to 0.545) | 0.486 (0.480 to 0.491) | 0.610 (0.602 to 0.618) | 0.549 (0.540 to 0.559) |
| **2018** | Opioids | 204679 | 213758 | 132149 | 364751 | 466631 | 309167 | 245890 |
| | Gabapentinoids | 132826 | 115943 | 123193 | 215541 | 247176 | 206896 | 149142 |
| | Gabapentinoid/Opioid ratio (95%CI) | 0.649 (0.647 to 0.654) | 0.542 (0.540 to 0.545) | 0.932 (0.913 to 0.952) | 0.591 (0.586 to 0.595) | 0.530 (0.528 to 0.532) | 0.669 (0.664 to 0.674) | 0.607 (0.601 to 0.612) |
| **2019** | Opioids | 201538 | 211603 | 133725 | 369234 | 461972 | 309545 | 242288 |
| | Gabapentinoids | 138967 | 117964 | 109657 | 229790 | 251415 | 215364 | 156517 |
| | Gabapentinoid/Opioid ratio (95%CI) | 0.690 (0.688 to 0.692) | 0.557 (0.556 to 0.559) | 0.821 (0.817 to 0.824) | 0.622 (0.619 to 0.625) | 0.544 (0.543 to 0.545) | 0.695 (0.693 to 0.698) | 0.646 (0.644 to 0.648) |
| **2020** | Opioids | 199369 | 210202 | 135237 | 370295 | 461678 | 308719 | 241610 |
| | Gabapentinoids | 142270 | 120299 | 107617 | 241681 | 255145 | 222747 | 162378 |
| | Gabapentinoid/Opioid ratio (95%CI) | 0.710 (0.710 to 0.711) | 0.570 (0.570 to 0.570) | 0.797 (0.796 to 0.797) | 0.651 (0.650 to 0.651) | 0.553 (0.552 to 0.553) | 0.720 (0.720 to 0.720) | 0.669 (0.668 to 0.669) |

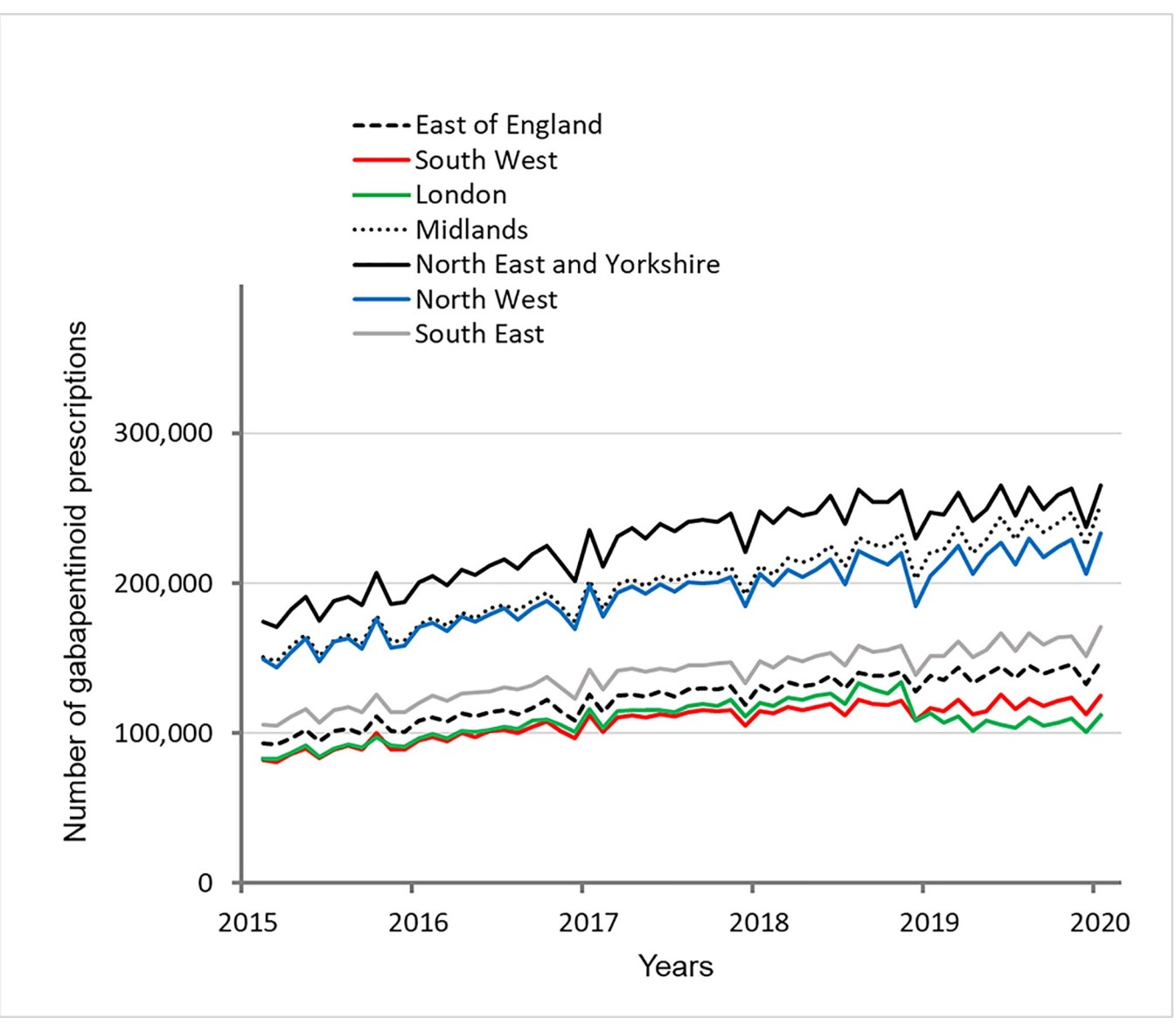

**Fig 2. Number of prescriptions for gabapentin and pregabalin in each region of England.**

## Discussion

### Prescription changes

This study describes the regional evolution in opioid and gabapentinoid prescriptions in England. The number of opioid prescriptions in various regions has essentially stabilized over time (ranging from -3.3% to +2.2%). Meanwhile, prescriptions of gabapentin and pregabalin increased between 2015 and 2020 (ranging from +1.5% to 11.9% every year). Additionally, we found that prescriptions for gabapentin and pregabalin declined in most areas by early 2019, after which the rate of prescription growth declined significantly but remained lower in 2020 only in the London commissioning region (by 12.3%).

In order to study the co-evolution of the prescriptions of gabapentinoids and opioids, we looked at the ratios between gabapentinoids and opioids. From 2015 to the end of 2018, there

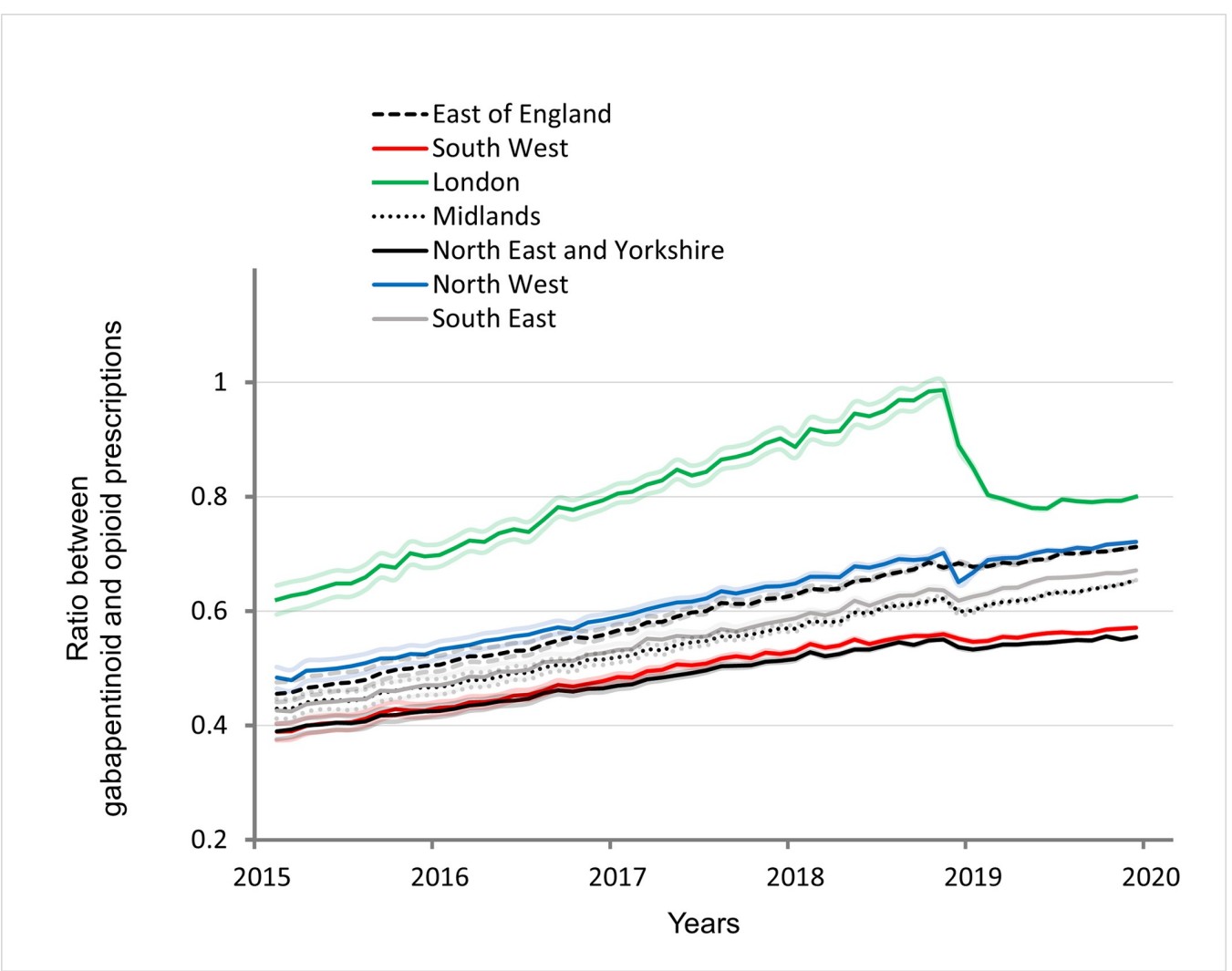

**Fig 3. Ratio between gabapentinoid and opioid prescriptions in each region of England.** The faded lines represent 95% confidence intervals calculated, for each month, on a rolling year.

was a marked increase in all regions, especially in the London commissioning region. At the beginning of 2019, the ratio of the different regions fell, after which the growth rate remained at a low level, but only significantly lower in the London commissioning (-14.4%; 95%CI:-12.8 to -16.3).

## Implications in terms of societal impact and national policy

As the public became aware of the potential risk of opioid use disorder, medical staff began to reconsider pain management and seek more appropriate medication prescriptions, including non-opioid pain relievers [4, 5]. Among other things to reduce opioid dependence, gabapentin and pregabalin have been gradually introduced. Between 2007 and 2017, the proportion of patients receiving primary opioid analgesics in combination with prescriptions of gabapentinoids or opioid analgesics in combination with prescriptions of benzodiazepines in primary care may have tripled approximately [16]. In some series, 20 to 25% of patients who started treatment with gabapentinoids were prescribed opioids [17]. If the goal were to partially

prevent opioid tolerance and improve the quality of treatment, especially for neuropathic pain, this effect could have been achieved at the cost of increased side effects, and, at best, uncertain [14, 8–20]. In the treatment of non-neuropathic pain, the use of gabapentinoids in combination with opioid analgesics may reduce short-term opioid use and short-term pain levels, but compared to placebo, the incidence of drug-related adverse reactions is higher [21–23].

Although gabapentinoids were initially identified as unlikely to be abused, in recent years gabapentinoids use disorders has gradually become a public health problem that cannot be ignored [3, 8, 10, 11, 16]. In 2016, the UK's Advisory Council on the Misuse of Drugs recommended that pregabalin and gabapentin be controlled under the Misuse of Drugs Act 1971 as Class C substances and listed under the Misuse of Drugs Regulations 2001 as Schedule 3 [9]. In the UK, controlled drugs are divided into classes A, B and C, depending on potential harms. Class C drugs are considered the least harmful. Schedule 3 implies specific requirements for the prescribing, dispensing, recording and safe storage of certain drugs, such as certain benzodiazepines. In April 2019, pregabalin and gabapentin were classified as Class C substances and listed as additional substances under the Misuse of Drugs Regulations 2001 [24]. The reclassification decision is based on risk of gabapentinoid misuse, abuse and diversion [9]. The decision appears to be linked to the drop in prescriptions of gabapentin and pregabalin in early 2019 we observed in this study.

It was important for all of us to study these regional differences. A multi-regional analysis was essential to consider as some regions may have more or less success in limiting drug-related problems. Although the reasons for the variability observed (in the present study) are uncertain and beyond our analyses, these data open the way to the development and discussion of different strategies. The observed patterns may also warrant future work aimed at better explaining determinants and outcomes, especially those that are modifiable. Importantly, due to the data we had, only prescriptions from England (but dispensed in the UK), were analysed. It would make sense to supplement these analyses by data from other nations and countries and, at least in the UK, to study the effect of other major guidelines, such as the National Institute for Health and Care Excellence (NICE) guidelines on chronic pain, published in 2021 [25].

## Limitations

Despite the large amount of data representing the entire national prescription rate in England, this work has its limitations. The use of OpenPrescribing data implies that inherent limitations must be considered. These data analyses include primary care only and are not weighted by quantity or strength of items prescribed [15]. This is especially important given the change in gabapentinoid controlled drug scheduling during the study period, limiting the maximum number of days for each prescription. Most importantly, the data has been aggregated, excluding any analysis (and interpretation) at the patient level. This means that the ratios between gabapentinoid and opioid prescriptions have been observed at the regional level and cannot be automatically extrapolated to the patient level. In other words, it cannot be asserted that these data can be considered as a surrogate for the concomitant use of opioids and gabapentinoids. Prescribing patterns may have been different, depending on the practice or over time and it is not possible to describe what proportion of patients are prescribed both these drugs, as this is not available within the data. Importantly, comparisons between regions (for prescriptions numbers) are very difficult to interpret because the number of inhabitants is different too. However, the differences observed and the variability of the ratio were consistently observed and deserve to be underlined, particularly seeing their implications at a Public Health level.

## Conclusions

In England, the number of opioid prescriptions remained stable over the study period (2015 to 2020), but gabapentinoid prescriptions increased. The increase in the ratio between both suggests that gabapentinoids have not replaced opioids in the therapeutic armamentarium.

The decision to reclassify gabapentinoids as drugs at risk of misuse, abuse, and diversion appears to be linked (at least initially) to a decline in gabapentin and pregabalin prescriptions.

## Supporting information

**S1 Checklist. STROBE statement—checklist of items that should be included in reports of cohort studies.**
(DOC)

## Author Contributions

**Conceptualization:** Yixue Xia, Patrice Forget.

**Formal analysis:** Yixue Xia, Patrice Forget.

**Investigation:** Yixue Xia, Patrice Forget.

**Methodology:** Yixue Xia, Patrice Forget.

**Supervision:** Patrice Forget.

**Visualization:** Yixue Xia.

**Writing – original draft:** Yixue Xia.

**Writing – review & editing:** Patrice Forget.

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
