## [Decision Letter · Decision Letter 0]

20 Jul 2022

PONE-D-22-15405Opioid and gabapentinoid prescriptions in England from 2015 to 2020.PLOS ONE

Dear Dr. Forget,

Thank you for submitting your manuscript to PLOS ONE. After careful consideration, we feel that it has merit but does not fully meet PLOS ONE’s publication criteria as it currently stands. Therefore, we invite you to submit a revised version of the manuscript that addresses the points raised during the review process.

We look forward to receiving your revised manuscript.

Kind regards,

Jorge Enrique Machado-Alba, M.D; Ph.D

Academic Editor

PLOS ONE

Journal Requirements:

3. Please include a copy of Table 1 which you refer to in your text on page 13.

Reviewers' comments:

Reviewer's Responses to Questions

**Comments to the Author**

1. Is the manuscript technically sound, and do the data support the conclusions?

Reviewer #1: Partly

Reviewer #2: Partly

2. Has the statistical analysis been performed appropriately and rigorously? 

Reviewer #1: Yes

Reviewer #2: No

3. Have the authors made all data underlying the findings in their manuscript fully available?

Reviewer #1: No

Reviewer #2: Yes

4. Is the manuscript presented in an intelligible fashion and written in standard English?

Reviewer #1: No

Reviewer #2: Yes

5. Review Comments to the Author

Reviewer #1: This is an interesting subject, and worthy of further study.

However there are several issues which ideally need to be addressed before publication:

-The data from OpenPrescribing uses items. Although this may be acceptable, there is no mention of the limitations of this method, i.e. that it doesn't take into account quantity or strength of the items prescribed. This is particularly important given the change to the CD scheduling of gabapentinoids during the study period, limiting the maximum number of days which can be prescribed.

- I think it would be further helpful to more clearly describe the changes to the CD scheduling, and the effect it may have. The statement "The newly promulgated management plan likely influenced prescriptions rate" is unclear - I'm not aware of any management plans being put in place nationally. I'm assuming it relates to move to schedule 3, but this is not mentioned. As a side-note, i'm assuming the "Drug Abuse Act 1971" actually refers to the "Misuse of Drugs Act 1971", and the same for the Misuse of Drugs Regulations 2001.

- It is unclear from the text what the findings represent, due to the limitations of data. There is discussion in the introduction about the safety of the co-prescribed opioids and gabapentinoids. However it is not possible to describe what proportion of patients are prescribed both these drugs, as this is not available within the data. The limitations section on this should be strengthened.

- It would be helpful to see some discussion of demographic differences between regions, otherwise it's not clear why the authors have undertaken the analysis at regional level.

- It would be helpful to see further discussion about possible reasons for the change in ratio, e.g. due to changes in national guidance for other analgesic options.

- This statement is unclear: "about 1% of the general population is in a situation of drug abuse, a proportion that could reach 40% of people who receive prescriptions." This suggests that only 2.5% of patients receive prescriptions. Is this actually referring to patients receiving opioid or gabapentinoid prescriptions? Also, the research referenced does not appear to be a systematic review as described in the authors' text, but a relatively small study based in Appalachian Kentucky, and therefore clarification is needed, particularly given the ability to use this data in a very specific area with known drug abuse problems in the USA with relation to England, which has very different models of care.

- There are some points in the text where ideally the authors should revise the language in order to improve readability. In particular the conclusion should be reviewed for language.

Reviewer #2: Thank you for the opportunity to review this retrospective analysis of prescribing trends of opioids and gabapentinoids over time, in the UK. Please find some suggestions for improvements and/or clarifications.

Abstract: please provide more information about the data sources, and the analysis. There are no statistical results included in the abstract and no test of trend etc.. This should be included.

Please make sure the key points match the findings of the study

Introduction:

-Statement about the dangerous combination use of opioids and gabapentionoids should be stronger in my opinion with reference to at least 3 studies that point to increased mortality from the combo (e.g., refer to the 2 observational studies by Juulink et al; there are others).

-Please mention and reference risk of seizure from stopping abruptly

-Why was data presented as a ratio of opioids to gabapentinoids?

-Did the authors consider looking at opioid doses? Same question for gabapentin vs pregabalin. Why simply look at dispensations? These can change depending on whether prescriptions are for 1,2,4, 6 weeks etc…

- Any consideration to doing a time series analysis of this data?

- I see that costs are mentioned qualitatively in the results but not quantitatively. I also don’t see this in the methods. Should this be added?

-At times I see mention of prescriptions for individual classes and other times I see this reported as a ratio. This needs to be clarified and unified.

-Addiction should be replaced with “use disorder”

-Caution against mentioning that gabapentinoids have been introduced to reduce opioid dependence as this is a hypothesis, only and has not been demonstrated to my knowledge.

-consider reviewing this reference for peri-operative lack of effect: https://pubs.asahq.org/anesthesiology/article/133/2/265/109137/Perioperative-Use-of-Gabapentinoids-for-the#

- Readers who are not from the UK will not know that the management plan is. It is mentioned in a couple of places but it isn’t clear what it consists of and how it is relevant.

- issue with the first sentence on page 11

- I see many pivotal studies addressing the lack of efficacy and the harms of gabapentinoids are missing from the references. Gingras et al., Verret et al., works by Juurink, RCTs showing no effect but increased risk of adverse events that were published in JAMA, NEJM etc…the references could be more robust for this study

- I like the figures but the statistical analysis plan is not that clear to be and the analysis is also not well described or robust. How about at time series analysis? And how about including some measure of the size of the population and the dose vs dispensations. Also not clear whether a ratio was calculated and where that is presented graphically and why it was done.

6. PLOS authors have the option to publish the peer review history of their article (what does this mean?). If published, this will include your full peer review and any attached files.

Reviewer #1: **Yes: **Richard Croker

Reviewer #2: No

---

## [Author Response · Author response to Decision Letter 0]

30 Jul 2022

Dear Editor, dear Reviewers, 

Thank you for the kind and constructive comments. You will find here a revised version of our work. We hope that this will encounter all your concerns. We remain, of course, open to any other suggestion.

Kind regards, 

Patrice Forget and Yixue Xia

Reviewer #1: This is an interesting subject, and worthy of further study.

Answer: Thank your for this encouraging comment.

Reviewer #1: However there are several issues which ideally need to be addressed before publication:

The data from OpenPrescribing uses items. Although this may be acceptable, there is no mention of the limitations of this method, i.e. that it doesn't take into account quantity or strength of the items prescribed. This is particularly important given the change to the CD scheduling of gabapentinoids during the study period, limiting the maximum number of days which can be prescribed.

Answer: The reviewer is obviously right, and we agree that it wasn't well detailed. We have now further detailed the limitations (discussion) section as follows: The use of OpenPrescribing data implies that inherent limitations must be considered. These data analyses include primary care only and are not weighted by quantity or strength of items prescribed. This is especially important given the change in gabapentinoid controlled drug scheduling during the study period, limiting the maximum number of days for each prescription.

Reviewer #1: I think it would be further helpful to more clearly describe the changes to the CD scheduling, and the effect it may have. The statement "The newly promulgated management plan likely influenced prescriptions rate" is unclear - I'm not aware of any management plans being put in place nationally. I'm assuming it relates to move to schedule 3, but this is not mentioned. As a side-note, i'm assuming the "Drug Abuse Act 1971" actually refers to the "Misuse of Drugs Act 1971", and the same for the Misuse of Drugs Regulations 2001.

Answer: Thank you for spotting these sources of confusion. We agree this is unclear, especially for the non-UK reader. We have now rephrased and clarified this paragraph as follows: In 2016, the UK's Advisory Council on the Misuse of Drugs recommended that pregabalin and gabapentin be controlled under the Misuse of Drugs Act 1971 as Class C substances and listed under the Misuse of Drugs Regulations 2001 as Schedule 3. In the UK, controlled drugs are divided into classes A, B and C, depending on potential harms. Class C drugs are considered the least harmful. Schedule 3 implies specific requirements for the prescribing, dispensing, recording and safe storage of certain drugs, such as certain benzodiazepines. In April 2019, pregabalin and gabapentin were classified as Class C substances and listed as additional substances under the Misuse of Drugs Regulations 2001 (20). The reclassification decision is based on risk of gabapentinoid misuse, abuse and diversion. The plan could be closely linked to the drop in prescriptions of gabapentin and pregabalin in early 2019.

Reviewer #1: It is unclear from the text what the findings represent, due to the limitations of data. There is discussion in the introduction about the safety of the co-prescribed opioids and gabapentinoids. However it is not possible to describe what proportion of patients are prescribed both these drugs, as this is not available within the data. The limitations section on this should be strengthened.

Answer: We agree with this point. We have accordingly modified the following paragraph: Prescribing patterns may have been different, depending on the practice or over time and it is not possible to describe what proportion of patients are prescribed both these drugs, as this is not available within the data. /…/ However, the differences observed and the variability of the ratio were consistently observed and deserve to be underlined, particularly seeing their implications at a Public Health level.

Reviewer #1: It would be helpful to see some discussion of demographic differences between regions, otherwise it's not clear why the authors have undertaken the analysis at regional level.

Answer: This is right, even if beyond our analyses. We tried not to be too speculative and mentioned regional/national differences at the end of the introduction, to introduce it from the first section. We also added in the discussion: It was important for all of us to study these regional differences. A multi-regional analysis was essential to consider as some regions may have more or less success in limiting drug-related problems. Although the reasons for the variability observed (in the present study) are uncertain and beyond our analyses, these data open the way to the development and discussion of different strategies. The observed patterns may also warrant future work aimed at better explaining determinants and outcomes, especially those that are modifiable.

Reviewer #1: It would be helpful to see further discussion about possible reasons for the change in ratio, e.g. due to changes in national guidance for other analgesic options.

Answer: To complement the answer to the previous question, we have also added the following sentence: Importantly, due to the data we had, only prescriptions from England (but dispensed in the UK), were analysed. It would make sense to supplement these analyses by data from other nations and countries and, at least in the UK, to study the effect of other major guidelines, such as the National Institute for Health and Care Excellence (NICE) guidelines on chronic pain, published in 2021 (20).

Reviewer #1: This statement is unclear: "about 1% of the general population is in a situation of drug abuse, a proportion that could reach 40% of people who receive prescriptions." This suggests that only 2.5% of patients receive prescriptions. Is this actually referring to patients receiving opioid or gabapentinoid prescriptions? Also, the research referenced does not appear to be a systematic review as described in the authors' text, but a relatively small study based in Appalachian Kentucky, and therefore clarification is needed, particularly given the ability to use this data in a very specific area with known drug abuse problems in the USA with relation to England, which has very different models of care.

Answer: The reviewer is correct, and we have chosen to remove the sentence, as even weaker statements could still be misinterpreted, as this cannot be generalized in any way.

Reviewer #1: There are some points in the text where ideally the authors should revise the language in order to improve readability. In particular the conclusion should be reviewed for language.

Answer: We have completely revised the text for typos. We rewrote the conclusion as follows: Gabapentinoids are increasingly prescribed in England. The number of opioid prescriptions remained stable over the study period (2015 to 2020), but the increase in the ratio suggests that gabapentinoids have not replaced opioids in the therapeutic armamentarium. The decision to reclassify gabapentinoids as drugs at risk of misuse, abuse, and diversion appears to be linked (at least initially) to a decline in gabapentin and pregabalin prescriptions.

We would like to thank the reviewer, and especially because the constructive comments reassure us that much can still be done to improve knowledge on this subject.

---

Reviewer #2: Thank you for the opportunity to review this retrospective analysis of prescribing trends of opioids and gabapentinoids over time, in the UK. Please find some suggestions for improvements and/or clarifications.

Abstract: please provide more information about the data sources, and the analysis. There are no statistical results included in the abstract and no test of trend etc.. This should be included.

Answer: This is correct. We have tried to complete is but to keep is brief as well. Here is the new abstract version: 

Results: During this period, opioid prescriptions remained stable (from -3.3% to +2.2%/year) and increased for gabapentinoids generally (from +1.5% to +2.2%). The ratio between opioid to gabapentinoid prescriptions increased by more than 20% in 2020 compared to 2015, variably between regions (F(6,406)=[120.2]; P<0.001; LSD Test: P<0.001; ANOVA for repeated measures: P<0.05). In 2019, a decline in the ratio occurred in all regions, but only persisting in the London commissioning region in 2020 (-14.4% in comparison with 2018, 95%CI:-12.8 to -16.3).

Conclusions: Gabapentinoids are increasingly prescribed in England. The ratio of gabapentinoid to opioid prescriptions in England increased from 2015 to 2020. The reclassification of gabapentinoids as controlled drugs, in 2019, may have been associated with a significant reduction, although larger prescribers may have been less influenced.

Reviewer #2: Please make sure the key points match the findings of the study

Answer: We checked the first three points to highlight, if required by the editor, and rewrote the last point as follows: In early 2019, the reclassification of gabapentinoids as controlled drugs may have been associated with a significant reduction, although larger prescribers may have been less influenced.

Reviewer #2: Introduction: Statement about the dangerous combination use of opioids and gabapentionoids should be stronger in my opinion with reference to at least 3 studies that point to increased mortality from the combo (e.g., refer to the 2 observational studies by Juulink et al; there are others).

Answer: We thank the reviewer for the suggestions, making it possible to highlight regional differences possibly linked to concomitant prescriptions of opioids and gabapentinoids. We have rewritten the paragraph as follows:

Additionally, the combined use of gabapentinoids and opioids appears to be associated with specific risks. In patients being prescribed opioids and gabapentinoids concomitantly, there is a substantial increase in the risk of opioid-related death (7,8). This concomitant use has been accused of being responsible of differences in opioid-related death rates in different parts of the UK (9).

References

8. Gomes T, Juurlink DN, Antoniou T, Mamdani MM, Paterson JM, van den Brink W. Gabapentin, opioids, and the risk of opioid-related death: A population-based nested case-control study. PLoS Med. 2017 Oct 3;14(10):e1002396.

9. Macleod J, Steer C, Tilling K, Cornish R, Marsden J, Millar T, Strang J, Hickman M. Prescription of benzodiazepines, z-drugs, and gabapentinoids and mortality risk in people receiving opioid agonist treatment: Observational study based on the UK Clinical Practice Research Datalink and Office for National Statistics death records. PLoS Med. 2019 Nov 26;16(11):e1002965. 

9. van Amsterdam J, van den Brink W, Pierce M. Explaining the Differences in Opioid Overdose Deaths between Scotland and England/Wales: Implications for European Opioid Policies. Eur Addict Res. 2021;27(6):399-412.

Reviewer #2: Please mention and reference risk of seizure from stopping abruptly

Answer: This has been described indeed. Now, the introduction contains the following sentence: 

Even if largely uncertain, it was hypothesised by some that gabapentinoids may help reduce opioids, prevent opioid tolerance, improve the quality of opioid analgesic therapy, and treat anxiety (3). However, these hypotheses have been, at least partially, rejected (4). Moreover, in high doses, patients may experience euphoria with gabapentinoids, and withdrawal after abruptly stopping use, including seizures (5,6). 

References

4. Verret M, Lauzier F, Zarychanski R, Perron C, Savard X, Pinard AM, Leblanc G, Cossi MJ, Neveu X, Turgeon AF; Canadian Perioperative Anesthesia Clinical Trials (PACT) Group. Perioperative Use of Gabapentinoids for the Management of Postoperative Acute Pain: A Systematic Review and Meta-analysis. Anesthesiology. 2020 Aug;133(2):265-279.

5. Goodman CW, Brett AS. Gabapentin and Pregabalin for Pain - Is Increased Prescribing a Cause for Concern? N Engl J Med. 2017 Aug 3;377(5):411-414. 

6. Schifano F, D'Offizi S, Piccione M, Corazza O, Deluca P, Davey Z, Di Melchiorre G, Di Furia L, Farré M, Flesland L, Mannonen M, Majava A, Pagani S, Peltoniemi T, Siemann H, Skutle A, Torrens M, Pezzolesi C, van der Kreeft P, Scherbaum N. Is there a recreational misuse potential for pregabalin? Analysis of anecdotal online reports in comparison with related gabapentin and clonazepam data. Psychother Psychosom. 2011;80(2):118-22. 

4. Mersfelder TL, Nichols WH. Gabapentin: Abuse, Dependence, and Withdrawal. Ann Pharmacother. 2016 Mar;50(3):229-33. 

Reviewer #2: Why was data presented as a ratio of opioids to gabapentinoids?

Answer: We looked not only at opioid prescriptions and gabapentinoid prescriptions, but also at concomitant use. At the practice and population level, we may have observed different or similar, but simultaneous changes in the prescribing patterns of the two drug classes. These simultaneous changes could have been invisible if they had not been highlighted by ratio analyses. Indeed, we identified in this study differences in ratios, not evident in the prescription data analyzed separately. This is particularly clear when looking at the difference seen in 2019. We have made this clear in various places in the summary and the main text.

Reviewer #2: Did the authors consider looking at opioid doses? Same question for gabapentin vs pregabalin. Why simply look at dispensations? These can change depending on whether prescriptions are for 1,2,4, 6 weeks etc…

Answer: As pointed out by Reviewer #1, OpenPrescribing data analyses are limited by their content. This is why we did not performed these additional analyses. We have not detailed these limitations as follows: The use of OpenPrescribing data implies that inherent limitations must be considered. These data analyses include primary care only and are not weighted by quantity or strength of items prescribed. This is especially important given the change in gabapentinoid controlled drug scheduling during the study period, limiting the maximum number of days for each prescription.

Reviewer #2: Any consideration to doing a time series analysis of this data?

Answer: This is an excellent suggestion. We did not plan to do a time series analysis, and, for this, should hypothesise that these data may have an internal structure (such as autocorrelation, trend or seasonal variation). Of course, this is partly what we did. However, as we think that the main driver might be modifiable (at a prescriber level), we believe descriptive analyses were the most important aspects we were interested in.

Reviewer #2: I see that costs are mentioned qualitatively in the results but not quantitatively. I also don’t see this in the methods. Should this be added?

Answer: We agree that this might be interesting for some, although, in this case, consequences of prescriptions may be important to consider as well. This is why we weren’t convinced by a direct added value of these isolated data. We are, in fact, really afraid that any firm conclusion would be mostly speculative and would weaken the entire work. However, we have precised now that: Costs were explored while considering all prescriptions as a whole, even if a pharmaco-economic study is beyond the purpose of this work.

Reviewer #2: At times I see mention of prescriptions for individual classes and other times I see this reported as a ratio. This needs to be clarified and unified.

Answer: We have checked that the presentation of results is as systematic as possible and corrected it according to the following order; 1. Opioids (absolute); 2. Gabapentinoids; 3. Ratios.

Reviewer #2: Addiction should be replaced with “use disorder”

Answer: Correct. We have also checked and replaced all the places were abuse was to be corrected too.

Reviewer #2: Caution against mentioning that gabapentinoids have been introduced to reduce opioid dependence as this is a hypothesis, only and has not been demonstrated to my knowledge.

Answer: The reviewer is right. We have rephrased it much more cautiously: Even if largely uncertain, it was hypothesised by some that gabapentinoids may help reduce opioids, prevent opioid tolerance, improve the quality of opioid analgesic therapy, and treat anxiety (3).

Reviewer #2: Consider reviewing this reference for peri-operative lack of effect: https://pubs.asahq.org/anesthesiology/article/133/2/265/109137/Perioperative-Use-of-Gabapentinoids-for-the#

Answer: We thank the reviewer for highlighting this work. This well justifies, at least some of our current concerns and the risk of overreliance on gabapentinoids. We have added this reference in the introduction: However, these hypotheses have been, at least partially, rejected (4,5).

4. Verret M, Lauzier F, Zarychanski R, Perron C, Savard X, Pinard AM, Leblanc G, Cossi MJ, Neveu X, Turgeon AF; Canadian Perioperative Anesthesia Clinical Trials (PACT) Group. Perioperative Use of Gabapentinoids for the Management of Postoperative Acute Pain: A Systematic Review and Meta-analysis. Anesthesiology. 2020 Aug;133(2):265-279.

5. Goodman CW, Brett AS. Gabapentin and Pregabalin for Pain - Is Increased Prescribing a Cause for Concern? N Engl J Med. 2017 Aug 3;377(5):411-414.

Reviewer #2: Readers who are not from the UK will not know that the management plan is. It is mentioned in a couple of places but it isn’t clear what it consists of and how it is relevant.

Answer: That's absolutely correct. Throughout the manuscript, and according to the first reviewer, we have reworded and detailed to ensure that it would be clear and relevant for non-UK readers. Among other things, here is an added explanatory paragraph (re)written for discussion:

In 2016, the UK's Advisory Council on the Misuse of Drugs recommended that pregabalin and gabapentin be controlled under the Misuse of Drugs Act 1971 as Class C substances and listed under the Misuse of Drugs Regulations 2001 as Schedule 3. In the UK, controlled drugs are divided into classes A, B and C, depending on potential harms. Class C drugs are considered the least harmful. Schedule 3 implies specific requirements for the prescribing, dispensing, recording and safe storage of certain drugs, such as certain benzodiazepines. In April 2019, pregabalin and gabapentin were classified as Class C substances and listed as additional substances under the Misuse of Drugs Regulations 2001 (19).

Reviewer #2: Issue with the first sentence on page 11

Answer: Thank you. We have corrected the sentence.

Reviewer #2: I see many pivotal studies addressing the lack of efficacy and the harms of gabapentinoids are missing from the references. Gingras et al., Verret et al., works by Juurink, RCTs showing no effect but increased risk of adverse events that were published in JAMA, NEJM etc…the references could be more robust for this study

Answer: Thank you for this suggestion. According to this relevant comment, we tried to identify impactful publications, but without over-referencing. We have added these and hope this will address the reviewer’s concerns.

Reviewer #2: I like the figures but the statistical analysis plan is not that clear to be and the analysis is also not well described or robust. How about at time series analysis? And how about including some measure of the size of the population and the dose vs dispensations. Also not clear whether a ratio was calculated and where that is presented graphically and why it was done.

Answer: We agree that the statistical analysis part was not clear, and we have also reformatted the results section, both in the main manuscript and the abstract. 

To be concise, the reviewer will find here the abstracts results section, but will find many more details in the manuscript: During this period, opioid prescriptions remained stable (from -3.3% to +2.2%/year) and increased for gabapentinoids generally (from +1.5% to +2.2%). The ratio between opioid to gabapentinoid prescriptions increased by more than 20% in 2020 compared to 2015, variably between regions (F(6,406)=[120.2]; P<0.001; LSD Test: P<0.001; ANOVA for repeated measures: P<0.05). In 2019, a decline in the ratio occurred in all regions, but only persisting in the London commissioning region in 2020 (-14.4% in comparison with 2018, 95%CI:-12.8 to -16.3).

As explained above, we appreciate the suggestion of performing a time series analysis. However, not hypothesising that that these data may have an internal structure (such as autocorrelation, trend or seasonal variation), we did not perform it. We hope that the reviewer will find that acceptable, wishing to focus on the descriptive analyses, being the most important aspects to us.

We also have found and corrected an error in the Y axis of the figure 3 (initially presented multiplied by a factor 1000 to simplify the presentation). We apologise for this.

---

## [Decision Letter · Decision Letter 1]

27 Sep 2022

PONE-D-22-15405R1Opioid and gabapentinoid prescriptions in England from 2015 to 2020.PLOS ONE

Dear Dr. Forget,

Thank you for submitting your manuscript to PLOS ONE. After careful consideration, we feel that it has merit but does not fully meet PLOS ONE’s publication criteria as it currently stands. Therefore, we invite you to submit a revised version of the manuscript that addresses the points raised during the review process.

We look forward to receiving your revised manuscript.

Kind regards,

Jorge Enrique Machado-Alba, M.D; Ph.D

Academic Editor

PLOS ONE

Reviewers' comments:

Reviewer's Responses to Questions

**Comments to the Author**

1. If the authors have adequately addressed your comments raised in a previous round of review and you feel that this manuscript is now acceptable for publication, you may indicate that here to bypass the “Comments to the Author” section, enter your conflict of interest statement in the “Confidential to Editor” section, and submit your "Accept" recommendation.

Reviewer #1: (No Response)

Reviewer #2: All comments have been addressed

2. Is the manuscript technically sound, and do the data support the conclusions?

Reviewer #1: Partly

Reviewer #2: Yes

3. Has the statistical analysis been performed appropriately and rigorously? 

Reviewer #1: Yes

Reviewer #2: Yes

4. Have the authors made all data underlying the findings in their manuscript fully available?

Reviewer #1: No

Reviewer #2: Yes

5. Is the manuscript presented in an intelligible fashion and written in standard English?

Reviewer #1: Yes

Reviewer #2: Yes

6. Review Comments to the Author

Reviewer #1: Thank you for the revision to this paper.

However, I still have concerns regarding the basis for the analyses. As stated in my previous review, it is not possible to tell from the prescribing data available at OpenPrescribing.net whether a patient is taking opioids and gabapentinoids concomitantly, and that the limitations should be strengthened. The paper now states "However,

the differences observed and the variability of the ratio were consistently observed and deserve to be underlined, particularly seeing their implications at a Public Health level", and now includes "we described the evolution of the prescriptions of opioids and gabapentinoids, and calculated their ratios (as a surrogate for their concomitant use at a population level) for each month." I do not believe you can use this data to show any sort of surrogate of concomitant taking of opioid and gabapentinoid, for reasons stated above, and therefore this should be revised to remove this suggestion.

Reviewer #2: (No Response)

7. PLOS authors have the option to publish the peer review history of their article (what does this mean?). If published, this will include your full peer review and any attached files.

Reviewer #1: **Yes: **Richard Croker

Reviewer #2: No

---

## [Author Response · Author response to Decision Letter 1]

27 Sep 2022

Dear Editor, dear Reviewers, 

Thank you for the kind and constructive comment. You will find here a revised version of our work. We hope that this will encounter all your concerns. We remain, of course, open to any other suggestion.

Kind regards, 

Patrice Forget and Yixue Xia

Reviewer #1: Thank you for the revision to this paper.

However, I still have concerns regarding the basis for the analyses. As stated in my previous review, it is not possible to tell from the prescribing data available at OpenPrescribing.net whether a patient is taking opioids and gabapentinoids concomitantly, and that the limitations should be strengthened. 

The paper now states "However, the differences observed and the variability of the ratio were consistently observed and deserve to be underlined, particularly seeing their implications at a Public Health level", and now includes "we described the evolution of the prescriptions of opioids and gabapentinoids, and calculated their ratios (as a surrogate for their concomitant use at a population level) for each month." 

I do not believe you can use this data to show any sort of surrogate of concomitant taking of opioid and gabapentinoid, for reasons stated above, and therefore this should be revised to remove this suggestion.

Answer: Thank you for the comment. This is right, and what we may believe is different than what we can state. In other words, we fully agree with the reviewer that this is factually not supported by the data presented and can in no way be taken as certain. Accordingly, we have checked, throughout, and removed any mention of "surrogate" or “concomitant” and strengthened the limitation section as follows:

This means that the ratios between gabapentinoid and opioid prescriptions have been observed at the regional level and cannot be automatically extrapolated to the patient level. In other words, it cannot be asserted that these data can be considered as a surrogate for the concomitant use of opioids and gabapentinoids. Prescribing patterns may have been different, depending on the practice or over time and it is not possible to describe what proportion of patients are prescribed both these drugs, as this is not available within the data.

---

## [Decision Letter · Decision Letter 2]

17 Oct 2022

Opioid and gabapentinoid prescriptions in England from 2015 to 2020.

PONE-D-22-15405R2

Dear Dr. Forget,

We’re pleased to inform you that your manuscript has been judged scientifically suitable for publication and will be formally accepted for publication once it meets all outstanding technical requirements.

Kind regards,

Vijayaprakash Suppiah, PhD

Academic Editor

PLOS ONE

Reviewers' comments:

Reviewer's Responses to Questions

**Comments to the Author**

1. If the authors have adequately addressed your comments raised in a previous round of review and you feel that this manuscript is now acceptable for publication, you may indicate that here to bypass the “Comments to the Author” section, enter your conflict of interest statement in the “Confidential to Editor” section, and submit your "Accept" recommendation.

Reviewer #2: All comments have been addressed

2. Is the manuscript technically sound, and do the data support the conclusions?

Reviewer #2: Yes

3. Has the statistical analysis been performed appropriately and rigorously? 

Reviewer #2: Yes

4. Have the authors made all data underlying the findings in their manuscript fully available?

Reviewer #2: Yes

5. Is the manuscript presented in an intelligible fashion and written in standard English?

Reviewer #2: Yes

6. Review Comments to the Author

Reviewer #2: I didn't have any additional comments for this article. It was another reviewer who did. I think their response to the other reviewer is adequate.

7. PLOS authors have the option to publish the peer review history of their article (what does this mean?). If published, this will include your full peer review and any attached files.

Reviewer #2: No
